



# A systematic examination of the relationship between CDOM and
# DOC for various inland waters across China
Kaishan Song[1], Ying Zhao[1,2], Zhidan Wen[1], Jianhang Ma[1,2], Tiantian Shao[1],
Chong Fang[1,2], Yingxin Shang[1]
[1]Northeast Institute of Geography and Agroecology, CAS, Changchun, 130102, China
[2] University of Chinese Academy of Sciences, Beijing 100049, China
Corresponding author's E-mail: songks@iga.ac.cn; Tel: 86-431-85542364
**Abstract:** Chromophoric dissolved organic matter (CDOM) plays a vital role in
aquatic ecosystems. Strong relationship has been proven between CDOM and
dissolved organic carbon (DOC), which set the basis for remote estimation of DOC
with remote sensing data. An algorithm has been developed to retrieve DOC via
CDOM absorption at 275 and 295 nm with coastal waters. However, the relationship
between DOC and $a_{CDOM}(275)$ and $a_{CDOM}(295)$ for different types of inland waters are
still not clear. Further, is the relationship stable with different types of inland waters?
In the current investigation, samples from fresh lakes, saline lakes, rivers or streams,
urban water bodies, ice-covered lakes were examined. The regression model slopes
range from 1.03 for urban waters to 3.13 for river water, with extreme low slope value
for highly saline waters (slope is about 0.3); while coefficient of determinations ($R^2$)
range from 0.71 (urban waters) to 0.93 (winter waters). The specific CDOM
absorption (SUVA254) showed the similar trend, i.e., low values were observed for
saline water and waters from semi-arid or arid regions, where strong photo-bleaching



is expect due to thin ozone layers, less cloud cover, longer water residence time and
sunshine hours. In the contrast, high values were measured with waters developed in
wetlands or forest in Northeast China, where inverse environmental settings were
witnessed. The investigation also demonstrated that stronger relationships between
CDOM and DOC were revealed when CDOM275 were sorted by SUVA254
($0.78 < R^2 < 0.98$) or the ratio of CDOM250:CDOM365 ($0.78 < R^2 < 0.99$). Our results
highlight that non-unified relationship exhibits for different types of inland waters,
and remote sensing models for DOC need to be tuned with different inherent optical
parameters obtained from various types of waters for quantification of DOC in inland
waters.
**Keywords:** Absorption, CDOM, DOC, spectral slope, saline water, fresh water

## 1. Introduction

Inland waters play a substantial role for regulating climate at regional scale, and also
for global carbon cycling (Cole et al., 2007; Tranvik et al., 2009). Compared with
other terrestrial ecosystems, e.g., forest, grassland and agricultural ecosystem, inland
waters only occupy a small fraction (3.5%) of the earth surface (Verpoorter et al.,
2014). However, they play a disproportional role for global carbon cycling with
respect to carbon transportation, transformation and carbon storage (Tranvik et al.,
2009; Verpoorter et al., 2014). According to Tranvik et al. (2009), 2.9 Pg C/yr was
imported from terrestrial ecosystems to inland waters, of which about 0.6 Pg C was
buried in the lake sediment each year, 1.4 Pg C/yr was released into the air as $CO_2$ or



methane, and the rest of 0.9 Pg C/yr was exported to the ocean via river channels.
However, the amount of C retained in the inland waters is still not clear or the
uncertainty is still remained for the current knowledge (Raymond et al., 2013). It has
been proposed by several researchers that remote sensing might provide a promising
tool for quantification of various carbon fractions and carbon storage for inland waters
(Cole et al., 2007; Tranvik et al., 2009; Song et al., 2013; Kutser et al., 2015).

Colored dissolved organic matter (CDOM) is one of the largest bioactive

reservoirs at earth's surface (Para et al., 2010), and influences light transmittance in
aquatic ecosystems (Vodacek et al., 1997; Williamson and Rose, 2010). Dissolved
organic carbon (DOC), the major component of CDOM, is a source of nutrients and
energy for heterotrophic bacteria, and the mineralization of allochthonous DOC in the
aquatic systems into net source of $CO_2$ in the atmosphere (Jaffe et al., 2008; Raymond
et al., 2013). DOC also serves to mediate the chemical environment through
production of organic acids (Landon and Bishop, 2002; Brooks and Lemon, 2007),
enhance or alleviate toxicity of heavy metals (Cory et al., 2006). A bunch of
researches have been conducted to characterize the spatial and seasonal variations of
CDOM and DOC for both inland and oceanic waters (Vodacek et al., 1997; Neff et al.,
2006; Stedmon et al., 2011) in ice free season, but less is known about saline lakes
(Song et al., 2013; Wen et al., 2016), urban waters influenced by sewage effluent and
ice covered waters in winter (Belzile et al., 2000, 2002).

The relationship between DOC and CDOM sets a bridge for remote estimation of

DOC in both oceanic water (Hoge et al., 1996; Bricaud et al., 2012; Nelson et al.,



2012) and inland waters (Yu et al., 2010; Griffin et al., 2011; Song et al., 2013; Zhu et
al., 2013). Thus, various attempts have been made to examine the relationship
between DOC and CDOM. According to Fichot and and Benner (2011), close
relationship between CDOM and DOC was observed for water from Mexican Gulf,
and stable regression model was established between DOC and $a_{CDOM}(275)$ and
$a_{CDOM}(295)$. Similar findings also observed in other estuary waters along a salinity
gradient, e.g., the Baltic Sea along the Finish Gulf (Kowalczuk et al., 2006), the
Chesapeake Bay (Le et al., 2013). However, investigation by Chen et al. (2004) also
indicated that the relationship between CDOM and DOC was not conservative due to
some process could either be estuarine mixing or photo-degradation. Similar
arguments were raised by Spencer et al. (2009) for waters from Congo River and for
waters across the mainland of USA (Spencer et al., 2012). Jiang et al. (2012) also
examined the relationship between DOC and CDOM for Lake Taihu, and found that a
relative stable relationship was observed for water samples collected in different
seasons except those measured in winter. Further, obvious seasonal variations were
observed, which could be explained by the mixing of various endmembers of CDOM
originated from different types of terrestrial ecosystems and internal source as well
(Zhang et al., 2010; Spencer et al., 2012).
As argued by Tranvik et al. (2009) and Raymond et al. (2013), remote sensing
technology was supposed to play a vital role in quantification of inland waters for
carbon cycling. To date, various attempts have been made to characterize DOC and
CDOM for both oceanic and inland waters and assess the relationship between these



two forms of carbons (Vodacek et al., 1999; Fichot and Benner, 2011; Griffin et al.,
2011; Spencer et al., 2012; Zhu et al., 2013). However, the variation for this
relationship with various types of inland waters, especially saline water and urban
water bodies were not examined in depth. In this study, the characteristics of DOC
and CDOM within different types of inland waters across China were examined to
determine its spatial feature associated with landscape variations, hydrologic
conditions and saline gradients. To specific, the objectives of this study are to: 1)
examine the relationship between CDOM and DOC concentrations across a wide
range of waters with various physical, chemical and biological conditions, 2) compare
the behavior of the relationships between DOC and CDOM for various water types,
and 3) establish model for the relationship between DOC and CDOM based on the
sorted CDOM absorption features, e.g., SUVA254 and the ratio of $a_{250} : a_{365}$ (DeHaan,
1993; Weishaar et al., 2003). To address these objectives, 1504 water samples were
collected in fresh and saline water lakes, reservoirs, rivers and streams, ponds across
China that encompass a broad ranges of DOC, CDOM concentrations with various
natural conditions, e.g., temperature, precipitation, hydrology, morphology, soil type
and landscape gradients. The findings from this research is essential for understanding
the relationship between DOC and CDOM with various types of inland waters, which
set a bridge for remote estimation of DOC contained in lakes or reservoirs.
**2. Materials and Methods**
The dataset is composed of five subsets of samples collected from various types of
waters across China (Table 1), which encompassed a wide range of DOC and CDOM





originating from different sources. The first dataset (n = 288; from early spring 2009
to late October 2014) was measured from samples collected in fresh lakes and
reservoirs for describing variations in absorption properties of different CDOM and
DOC sources during the growing season with various landscape types. The second
dataset (n = 345; from early spring 2010 to late mid-September 2014) was measured
from samples collected in brackish to saline water bodies for investigating variations
in CDOM absorption properties and hydrological impact on DOC concentration. The
third dataset (n =322; from early May 2012 to late October 2014) was measured from
samples collected in rivers and streams across a wide region in China. The fourth data
(n = 328; from 2011 to 2014 in the ice frozen season) was measured from samples
collected in northeast China in winter from both lake ice and underlying waters. The
fifth dataset (n = 221; from early May 2013 to mid-October 2014) was measured of
samples from urban water bodies, including lakes, ponds, rivers and streams, which
was severely influenced by sewage effluents. It is expected that CDOM and DOC
from various water types may illustrate a general trend between these two parameters.

**[Insert Fig.1 about here]**


**2.1 Water quality determination**
In the laboratory, water salinity was measured through DDS-307 electrical
conductivity (EC) meter (μS/cm) at room temperature ($20\pm2\,℃$) and transformed to
practical salinity units (PSU). Water samples were filtered and extracted with acetone
for chlorophyll-a (Chl-a) concentration determination using a Shimadzu UV-2050PC




spectrophotometer (Song et al., 2013). Total suspended matter (TSM) was determined
gravimetrically, details can be found in Song et al. (2013). DOC concentrations were
determined by high temperature combustion (HTC) with water samples filtered
through pre-combusted 0.45 μm GF/F filters (Song et al., 2013). The standards for
dissolved total carbon (DTC) were prepared from reagent grade potassium hydrogen
phthalate in ultra-pure water, while dissolved inorganic carbon (DIC) were
determined using a mixture of anhydrous sodium carbonate and sodium hydrogen
carbonate. DOC was calculated by subtracting DIC from DTC, both of which were
measured by a Total Organic Carbon Analyzer (Shimadzu, TOC-VCPN). Total
nitrogen (TN) was measured based on the absorption levels at 146 nm of water
samples decomposed with alkaline potassium peroxydisulfate. Total phosphorus (TP)
was determined using the molybdenum blue method after the samples were digested
with potassium peroxydisulfate (APHA, 1998). A PHS-3C pH meter was used to
determine pH at room temperature (20±2 ℃) in laboratory.

**2.2 CDOM absorption and spectral slope (S) derivation**
First, all the samples were filtered at low pressure, first through a pre-combusted
Whatman GF/F filter (0.7μm) in the laboratory, and then the filtrate were further
filtered through pre-rinsed 25 mm Millipore membrane cellulose filter (0.22 μm) at a
low pressure. Absorption spectra were obtained between 200 and 800 nm at 1 nm
increment using a Shimadzu UV-2600PC UV-Vis (Shimadzu Inc., Japan) dual beam
spectrophotometer through a 1 cm quartz cuvette (or 5 cm cuvette for ice melted





water samples), and Milli-Q water was used as reference for CDOM absorption
measurements. The absorption coefficient ($a_{CDOM}$) was calculated from the measured
optical density ($OD$) of samples using Eq. (1):
$$a_{CDOM}(\lambda) = 2.303[OD_{S(\lambda)} - OD_{(null)}]/\beta \qquad (1)$$
where $\beta$ is the cuvette path length (0.01 or 0.05m) and 2.303 is the conversion factor
of base 10 to base $e$ logarithms. To remove the scattering effect from fine particles
remained in the filtered solutions, a necessitated correction was implemented. The
$OD_{(null)}$ is the average optical density over 740–750 nm, which is assumed to be zero
for the absorbance of CDOM (Zhang et al., 2007). All absorption measurements were
conducted within 48 h after the samples were shipped back to the laboratory. The
specific CDOM absorption coefficients were calculated as the ratio of $a_{CDOM}(\lambda)$
against DOC concentration, and denoted as $a^*_{CDOM}(\lambda)$ with units of ($m^{-1}.L.mg^{-1}$). In
the current study, the value of $a^*_{CDOM}(\lambda)$ at reference wavelength of 350 nm was
calculated as suggested by previous investigations (Vodacek et al., 1999; Fichot and
Benner, 2011), which will be further used as spectral index for establishing
relationship between CDOM and DOC.

A CDOM absorption spectrum, $a_{CDOM}(\lambda)$, is generally expressed as an

exponential function (Babin et al., 2003):
$$a_{CDOM}(\lambda_i) = a_{CDOM}(\lambda_r)e^{-S(\lambda_i - \lambda_r)} \qquad (2)$$
where $a_{CDOM}(\lambda_i)$ is the CDOM absorption at a given wavelength $\lambda_i$, $a_{CDOM}(\lambda_r)$ is the
absorption estimate at the reference wavelength (i.e., $\lambda_r = 440$ nm) and $S$ is the
spectral slope. The $S$ is calculated by fitting the data to a nonlinear model over a




wavelength range of 300 to 500 nm as suggested by Zhang et al. (2007).

## 3. Results and discussion

In all datasets collected over different types of water bodies across China, a large
diversity of inland waters with varying water qualities was encountered. High average
Chl-a concentrations ($46.44 \pm 59.71$ µgL$^{-1}$) are observed in these waters, which ranged
between 0.28-521.12 µg/L. As shown in Table 1, fresh water, saline water and
particularly urban water bodies all exhibited high TN and TP values, indicating that
most of the waters are highly eutrophic. It should be noted that even winter water
samples also revealed high Chl-a concentration ($7.3 \pm 19.7$ µgL$^{-1}$), which is resulted
from high TN ($4.3 \pm 5.4$ mgL$^{-1}$) and TP ($0.7 \pm 0.6$ mgL$^{-1}$) concentrations even under ice
covered conditions. Due to regional hydro-geologic and climatic conditions, most
waters in the semi-arid and arid regions have high electric conductivity (EC:
1067-41000 µs/cm) and pH values (7.1-11.4). Overall, waters are highly turbid by
showing high concentration of TSM ($119.55 \pm 131.37$ mgL$^{-1}$), but different water
types demonstrated obvious variations in the water column (Table 1). Hydrographic
conditions exert strong impact on water turbidity and TSM concentration, thus these
two parameters for river and stream samples were not measured in this study (Table 1).
Large variations of water quality parameters extensive geographic conditions set a
more representative basis for examination of the relationship between DOC and
CDOM, which is potentially helpful for remote estimate of DOC through CDOM
absorption properties (Kutser et al., 2015).

**[Insert Table 1 about here]**



### 3.1. DOC concentrations in various types of waters


The range of DOC concentrations spanned an order of magnitude over these waters
being investigated. As shown in Table 1, low averaged concentration of DOC was
observed for river waters, but even lower DOC concentrations were measured with
ice melting waters sampled in winter. It should be noted that large variations were
measured with waters from rivers and streams (Table 2). Generally, low DOC
concentrations were found in rivers or streams in the drainage systems developed in
Tibetan Plateau or arid regions where soil contains relative low concentration of soil
organic carbon, while inverse trend were found in rivers or streams surrounded by
forest or wetlands. Among the five types of waters investigated, high DOC
concentrations were recorded for saline waters, ranging from 2.3 to 300.6 mg/L. This
investigation indicated that saline waters originated from the Songnen Plain, the
HulunBuir Plateau and part of waters from Tibetan Plateau generally exhibits high
concentration of DOC, while some of waters supplied with snow melt water or ground
waters generally exhibit low DOC concentrations even with high salinity. Compared
with samples collected in growing seasons, higher DOC concentrations were observed
in ice covered water bodies (7.3-720 mg/L), which is due to the condensed effect
caused by the DOC expelled from ice formation (Bezilie et al., 2002). This condensed
effect is particularly marked for these shallow water bodies, where ice forming
remarkably condensed the DOC in the underlying waters (Zhao et al., 2016). As
shown in Table 2, even in river or saline water bodies, the concentrations of DOC
demonstrated obvious variations. Comparatively, river waters from Qinghai exhibited




lower DOC concentration, while these from Liaohe and Inner Mongolia showed much
higher concentration. Likewise, large variations were exhibited for saline waters of
different regions (Table 2). Saline waters from the Qinghai and Hulunbir showed
much higher DOC concentration, while these from the Xilinguole Plateau and the
Songnen Plain exhibited relative lower DOC concentrations.

**[Insert Table 2 about here]**

**3.2. DOC versus CDOM with various types of waters**
*3.2.1 Fresh waters*
The relationships between DOC and CDOM have been examined based on CDOM
absorption spectra at different wavelength (Fichot and Benner, 2011; Spencer et al.,
2012). As suggested by Fichot and Benner (2011), CDOM absorptions at 275 nm
(CDOM275) and 295 nm have stable performances for DOC estimates. As shown in
Fig.2a, a strong relationship ($R^2 = 0.85$) between DOC and CDOM275 was exhibited
with samples collected in fresh lakes and reservoirs. Regression analyses of the
dataset collected from different regions indicated that the slope values varied from
1.87 to 3.22. The results indicated that water samples from North China and East
China turned to have lower regression slope values, where lakes and reservoirs
generally ranged from mesotrophic to eutrophic status. Phytoplankton degradation
may contribute relative large portion of DOC in these water bodies (Zhang et al.,
2010). Comparatively, fresh water bodies from Northeast China revealed larger
regression slope values, and CDOM from these water bodies are surrounded by forest,
wetlands and grassland generally exhibit high proportion of colored fractions (Helms





et al., 2008). Further, soils in Northeast China are endorsed with high organic carbon,
which may also contribute high concentration of DOC and CDOM in waters from this
region (Jin et al., 2016). Compared with waters from East and South China, water
bodies in Northeast China show less algal bloom due to the low temperature, thus
autochthonous CDOM is less presented in waters from Northeast China (Song et al.,
2013; Zhao et al., 2016).
*3.2.2 Saline lakes*
As shown in Fig.2b, a strong relationship ($R^2 = 0.85$) between DOC and CDOM275
was demonstrated for saline waters. However, compared to fresh waters, much lower
regression slope value (slope = 1.28) was exhibited for saline waters. Similar to fresh
water bodies, the slope values for most saline waters exhibited large variations from
different regions, ranging from 0.67 to 2.47. As the extreme case, the slope value is
only 0.33 as demonstrated in the embedded diagram in Fig.2b. Our analyses indicated
that the saline waters from semi-arid or arid regions, e.g., west Songnen Plain (2.47),
Hulunbir Plateau and East Inner Mongolia Plateau (1.79) generally exhibit higher
regression slope values. Whereas, water bodies from the western part of Inner
Mongolia Plateau (1.13), the Tibetan Plateau (0.86) exhibited low slope values, and
the extreme low value was measured with the Lake Qinhai from Tibetan Plateau, and
lakes from Tarim Basin, where lakes experience long resident time and strong solar
radiation enhances the photo-bleaching effects (Spencer et al., 2012; Song et al., 2013;
Wen et al., 2016). Thereby, less colored portion of DOC was presented in water
bodies in semi-arid to arid regions, especially for these closed lakes with enhanced



photochemical processes resulting in lower regression slope value (Spencer et al.,
2012). The findings highlighted that remote sensing of DOC through CDOM
absorption algorithm for saline waters was remarkably different from fresh waters.
*3.2.3 Stream and rivers*
Though some of the samples scattered from the regression line (Fig.2c), close
relationship between DOC and CDOM275 was revealed for samples collected in
rivers and streams. Compared with the other water types (Fig.2), the highest
regression slope value (slope = 3.13) was exhibited with river and stream water
samples. Further regression analysis with sub-datasets measured with water samples
collected in different regions indicated that slope values presented large variability,
ranging from 1.84 to 8.41. The lower regression slope was recorded with water
samples collected in rivers and stream in semi-arid and arid regions, e.g., the Tibetan
Plateau, Mongolia Plateau and Tarim Basin, while the higher values were found with
samples collected in streams originated from wetland and forest in Northeast China.
Rivers and streams in North, East and South China generally exhibit intermediate
value, ranging from 2.5 to 4.2. In addition, large river water generally presented
relatively low slope value, streams, especially head water originating from forest and
wetland dominated regions show higher regression slope value, which is consistent
with the finding from Helm et al. (2008) and Spencer et al. (2012). In fact, landscape
pattern in a specific watershed, including soil organic carbon, may be important
factors governing the terrestrial DOC and CDOM characteristics in rivers and streams
encompassed in the watershed (Wilson and Xenopoulos, 2008; Jaffe et al., 2008).




### 3.2.4 Urban waters

Although close relationship between DOC and CDOM275 was revealed with urban
waters (Fig.2d, $R^2 = 0.71$), it is much scattered compared with other water types
(Fig.2), particularly with samples presenting DOC concentration less than 60 mg/L.
Similarly, very large variability of regression slope values was demonstrated, ranging
from 0.78 to 4.16. It is apparent that urban water bodies are severely affected by
human activities, particularly sewage, effluents and runoff from urban impervious
surface containing large amount of DOM and nutrient discharge into urban waters.
Elevated nutrients generally result in algal bloom for some of the urban water bodies
(Chl-a range: 1.0-521.1 μg/L; average: 38.9 μg/L). Thereby, DOC and CDOM derived
from phytoplankton may also contribute a portion that should not be neglected (Zhao
et al., 2016; Zhang et al., 2010). More or less affected by sewage effluent, the DOM
in urban waters is much complex than those from natural water bodies. Thus, a large
variation of the relationship between DOC and CDOM275 is expected with urban
waters.

### 3.2.5 Ice covered lakes and reservoirs

As demonstrated in Fig.2e, a closest relationship ($R^2 = 0.93$) between DOC and
CDOM275 was recorded with waters beneath ice covered lakes and reservoirs in
Northeast China. It was argued that the close relationship indicated the concurrent
processes taken place for DOC accumulation and CDOM biogeochemical activities
(Finlay et al., 2003; Stedmon et al., 2011). The strong positive correlations between
DOC and CDOM275 is probably due to ice formation condensed these two




parameters. The other possible explanation is that ice and snow cover shield out most
of the solar radiation that may cause a series of biochemical process for CDOM
contained in water, the inflows and direct rainfall over lakes or reservoirs also
diminished, thus causing limited effect on DOC concentration and CDOM
composition (Uusikiv et al., 2010; Belzile et al., 2002). Further, the autochthonous
DOC and CDOM for ice covered waters are also very limited due to the weak primary
production in winter (7.3 μg/L). Thus, much close relationship between DOC and
CDOM is expected for winter waters.
Comparatively, a loose correlation between DOC and CDOM275 was
demonstrated for ice melting waters (Fig.2f) are probably due to the ice/water depth
ratio, which cause variation of dissolved components expelled during ice formation.
The other reason is probably due to the biologically derived DOC in the ice matrix,
which could be varied due to the light and nutrient conditions (Arrigo et al., 2010;
Muller et al., 2011). Apparently, CDOM from ice melting waters were mainly
originated from maternal water during the ice formation, also from algal biological
processes (Stedmon et al., 2009; Arrigo et al., 2010). The DOC and CDOM
concentrations in maternal waters, and ice formation processes may cause the
variations for their relationship, thus the regression slopes varies. Similarly, snow
cover, and nutrients in the ice also cause the variation for biochemical processes, that
ultimately result in the relationship between DOC and CDOM may differ from
corresponding waters (Bezilie et al., 2002; Spencer et al., 2009). Interestingly, the
regression slopes for ice samples (slope = 1.35) and under lying water sample (slope =





1.27) are very close, which may also explain that the dominant components of CDOM
and DOC in the ice are from maternal underlying waters.
**[Insert Fig.2 about here]**
**3.3 DOC versus CDOM based on SUVA254 and M ($a_{250}$:$a_{365}$) values**
Through comparison of the relationships between DOC and CDOM275, it can be seen
that the regression slope vales exhibit large variability for various types of waters. The
underlying reasons may lies in the aromacity and colored fractions in DOC
component (Spencer et al., 2009, 2012). Since SUVA254 is an effective indicator to
characterize CDOM molecular weight, and is calculated by the ratio of CDOM
absorption at 254 nm to DOC (Weishaar et al., 2003), it may reflect the regression
slope value between DOC and CDOM absorption at 275 nm. As shown in Fig.3a, it is
obvious that SUVA254 presented high values for both fresh water bodies, and waters
from rivers or streams as well. Saline water and winter water samples show
intermediate SUVA254 values, while urban water and ice melting water show lower
values. The M value ($a_{250}$ : $a_{365}$) is another indicator to demonstrate the variation of
molecular weight and aromacity of CDOM components (Dehaan, 1993). As shown in
Fig.3b, fresh water, river and stream water, and urban water exhibit low values, which
indicated that larger aromacity dominant for these three types of waters, whereas
saline water, winter water and ice melting water show higher M values. Since,
SUVA254 and M values reveal molecular weight and aromacity, it might help to
estimate DOC through CDOM absorption based on SUVA254 and M threshold values





for various types of waters being investigated.
**[Insert Fig.3 about here]**
*3.3.1 Regression based on SUVA254 grouping*
Based on the threshold value for SUVA254, eight subsets of paired DOC and
CDOM275 were grouped. Figs.4a to 4f demonstrated the regressions between DOC
and CDOM275 with a SUVA254 increment of unity. Fig.4g and 4h exhibited the
cases with SUVA254 threshold larger than unity. Except the regression model with
SUVA254 less than one (Fig. 4a), better performances were achieved for regression
models based on SUVA254 thresholds between 2 to 6 (Figs.4b-4f). As shown in
Fig.4a-4h, as a whole, the regression slope values have strong links with SUVA254
values, i.e., slope values increased with SUVA254 increment except for SUVA254
ranging from 6 to 8. It is still not clear whether it is because some outliers or the
complex relationship between SUVA254 and DOC, further investigation is required to
figure out the underlying reasons. It can be seen that in most of cases, the regression
models performed much better based on SUVA254 thresholds. The less promising
cases were related with subset of data with lower and high SUVA254 values,
relatively larger variations at inner groups are expected thus outperformed by
regression models with intermediate SUVA254 values.
**[Insert Fig.4 about here]**
*3.3.1 Regression based on M value grouping*
Likewise, regression models between DOC and CDOM275 were established based on



M threshold values (Fig.5). A relative loose correlation between DOC and CDOM275
was revealed with dataset where M value was less than 5 (Fig.5a). It should be noted
that the highest regression slope value was achieved among different groups of subset
of data (Figs.5a-5h). The large range of M value (0<M<5.0) may explain the scattered
data pairs in Fig.5a; similar reason can be ascribed to the group with M value ranging
from 4 to 6 (Fig.5b). Better regression models were achieved with intermediate M
value groups (Figs.5c-5f), where regression slope values were close to each other
(ranging from 1.15 to 1.38) with high determination of coefficients ($R^2$> 0.88). With
increased M values, small regression slope values were obtained (Figs.5g-h). Loose
relationship between DOC and CDOM275 was obtained with relative low or high M
values (Fig.5g). However, very close relationship ($R^2 = 0.99$) was yielded with
extremely high M values (Fig.5h). It can be seen that most of samples are from these
presented in embedded diagram in Fig.2b, the limited water bodies in the group may
be explain this coincidently high R-square value. With more samples collected from
different water bodies in this extreme group, a loose relationship between DOC and
CDOM275 may be expected, which also needs future explorations.

As noted in Figs.5c-5f, close regression slope values were obtained, implicating

that a comprehensive regression model with intermediate M value groups may be
achieved. As expected, a promising regression model (the diagram was not shown)
between DOC and CDOM 275 was achieved (y = 1.269x + 6.55, $R^2$ = 0.925, N = 998,
p < 0.001) with pooled dataset presenting in Figs.5c to 5f. As shown in Fig.6a, a close
relationship between DOC and CDOM 275 was obtained with the pooled dataset (N =





395 1504) collected from different types of inland waters. However, it should be admitted

396 that the extremely high DOC samples may advantageously contribute the better

397 performance of the regression model. Thus, regression model was established without

398 these eight samples (DOC > 300 mg/L), still acceptable accuracy can be achieved

399 (Fig.6b, $R^2$ = 0.66, p < 0.01). In addition, regression model based on logarithm

400 transformed pool dataset was also established (Fig.6c, $R^2$ = 0.82, p < 0.01). It can be

401 seen that most of the paired data sitting close to the regression line except some

402 scattered ones. Based on the regression analysis on pooled dataset, it can be

403 concluded that it is possible to derive DOC concentration based on CDOM absorption

404 spectra, and the latter parameter can be estimated from remotely sensed data (Zhu et

405 al., 2011; Kuster et al., 2015).

406       **[Insert Fig.5 and Fig.6 about here]**

408 **4. Conclusion**

409 As a powerful means, remote sensing plays a crucial role in assessing CDOM and

410 DOC in lake and reservoir waters. However, in order to get accurate estimates of

411 CDOM and DOC in waters, it is necessary to get insight into the regional water

412 optical properties for developing semi-analytical or analytical models with remotely

413 sensed data. Based on CDOM absorption spectral measurements and DOC laboratory

414 analysis, we have investigated the relationships between CDOM and DOC for various

415 water types systematically. The investigation showed that CDOM absorption varied

416 significantly, and generally river waters and fresh lake waters exhibit high CDOM





absorption values and specific CDOM absorption (SUVA254) as well. On the contrast,
saline waters illustrate low SUVA254 values due to the long residence time and strong
photo-bleaching effects on waters in the semi-arid regions. Influenced by effluents
and sewage waters, CDOM from urban water bodies showed much complex
absorption feature. With respect to ice melting water samples, SUVA254 for CDOM
was lowest for all groups of waters concerned.

The current investigation indicated that the relationships between CDOM

absorption and DOC varied significantly by showing different slope values with
various water types of regression models. The slope values for saline and urban
waters are close to unity, while river water exhibited highest slope value (~ 3.1) of all
water types concerned, and other water types are in between. When all the data set
pooled together, the slope for regression model is about 1.3, but with much higher
uncertainty ($R^2$ = 0.66). Regression model accuracy for CDOM275 against DOC was
improved when CDOM absorptions were divided into different sub-groups according
to SUVA254 or M values ($a_{250}$:$a_{365}$). This finding highlights that remote sensing
models for DOC estimates based on the relationship between CDOM and DOC need
to consider water types or cluster waters into several groups according to their
absorption features, ultimately improved model accuracy is expected.

**Acknowledgements**
The authors would like to thank financial supports from Natural Science Foundation
of China (No.41471290), and "One Hundred Talents" Program from Chinese





Academy of Sciences granted to Dr. Kaishan Song. Thanks are also extended to all
the staff and master students for their efforts in field data collection and laboratory
analysis.

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



**Figures**
Fig.1. Water types and sample distributions across the mainland of China.

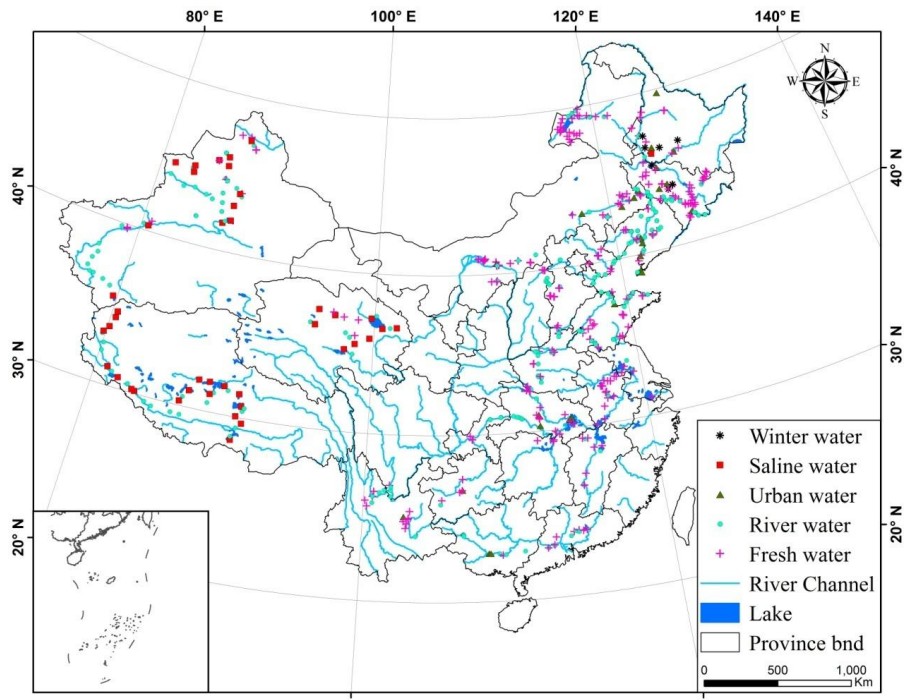




Fig.2. Fitting equations of DOC against CDOM275 for different types of inland
waters, (a) samples from fresh water lakes, (b) samples from saline water lakes, (c)
samples from river waters, (d) samples from urban waters, (e) samples from ice
covered waters, and (f) samples from ice melting waters.

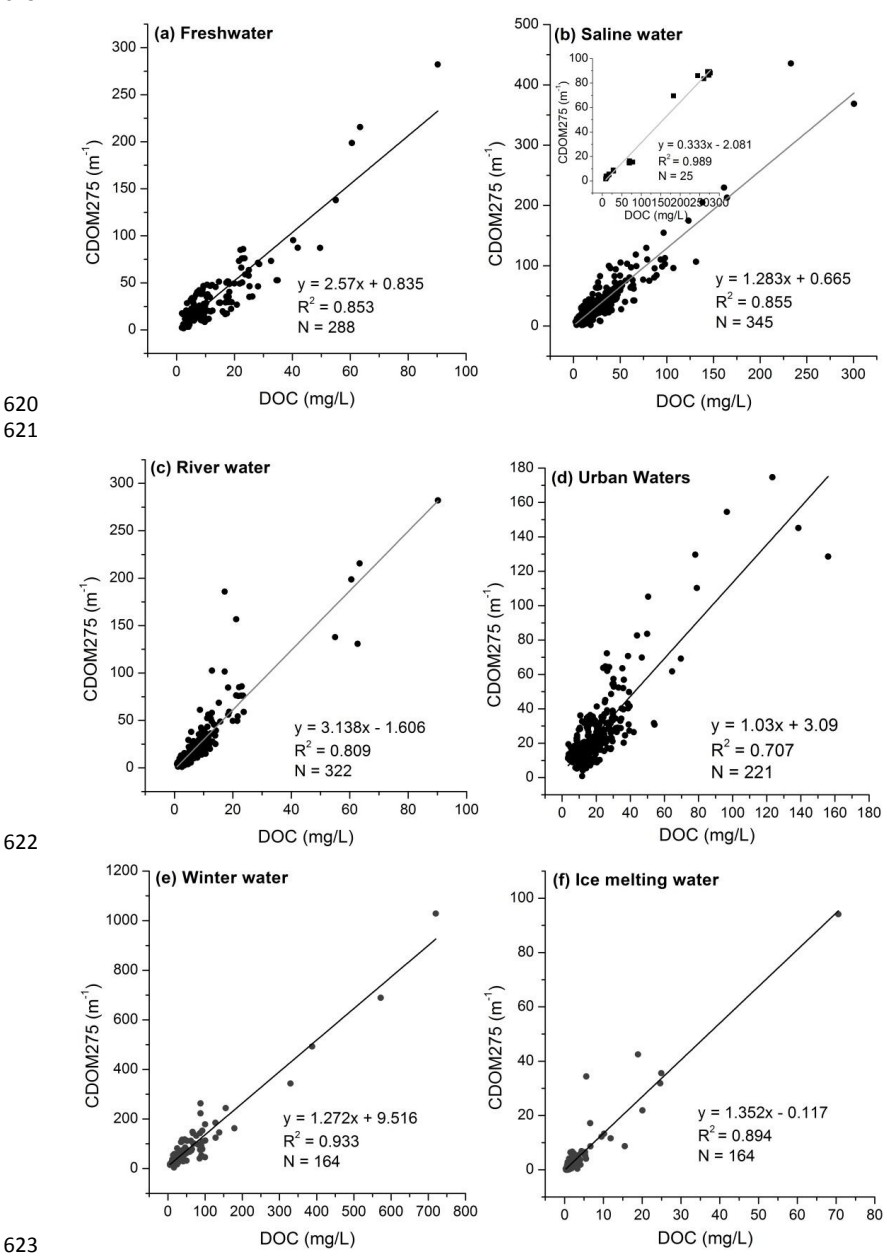






Fig.3. Comparison of (a) SUVA254, and (b) M ($a_{250}{:}a_{365}$) values for various types of
inland waters. FW, fresh lake water; SW, saline lake water, RW, river or stream water;
UW, urban water; WW, winter water from Northeast China; IMW, ice melt water from
Northeast China.

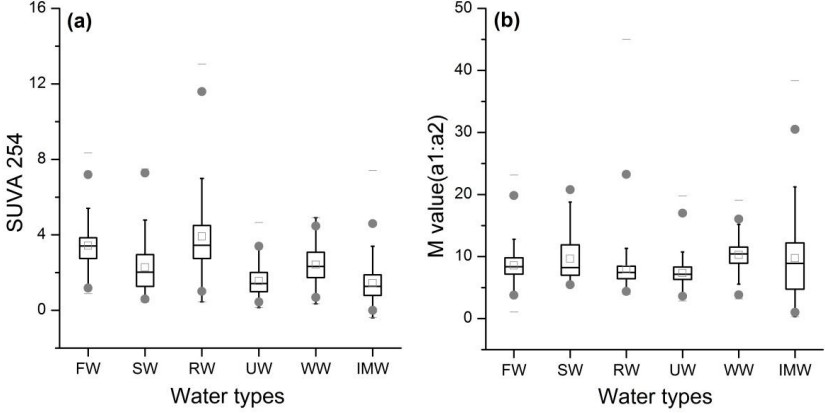




Fig.4. Regression models for DOC estimation via CDOM 275 sorted by SUVA254
ranges, (a) SUVA254 > 1.0, (b) 1.0 < SUVA254 < 2.0, (c) 2.0 < SUVA254 < 3.0, (d)
3.0 < SUVA254 < 4.0, (e) 4.0 < SUVA254 < 5.0, (f) 5.0 < SUVA254 < 6.0, (g) 6.0 <
SUVA254 < 8.0, and (h) 8.0 < SUVA254 < 13.1

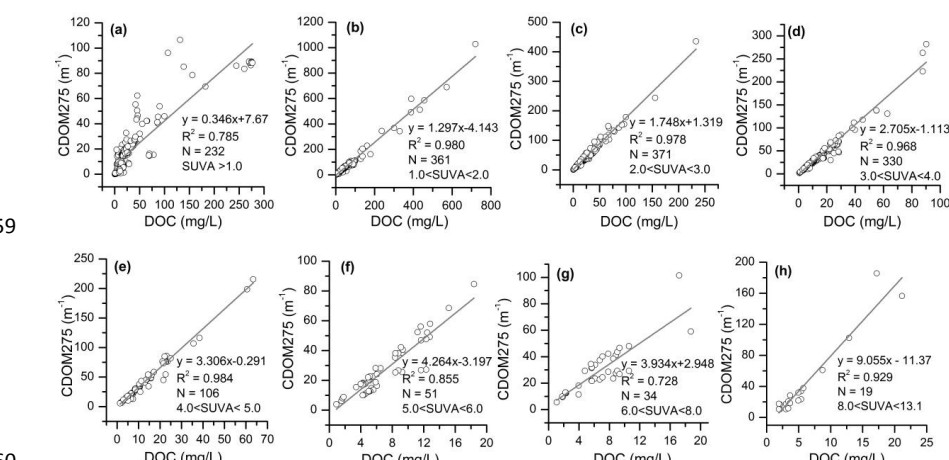







Fig.5. Regression models for DOC estimation via CDOM 275 sorted by M (a1:a2:
CDOM250:CDOM365) values ranges, (a) M < 5.0, (b) 4.0 < M< 6.0, (c) 6.0 < M<
6.0, (d) 7.0 < M< 8.0, (e) 8.0 < M< 10.0, (f) 10.0 < M< 12.0, (g) 12.0 < M< 20.0, and
(h) 20.0 < M< 68.0.

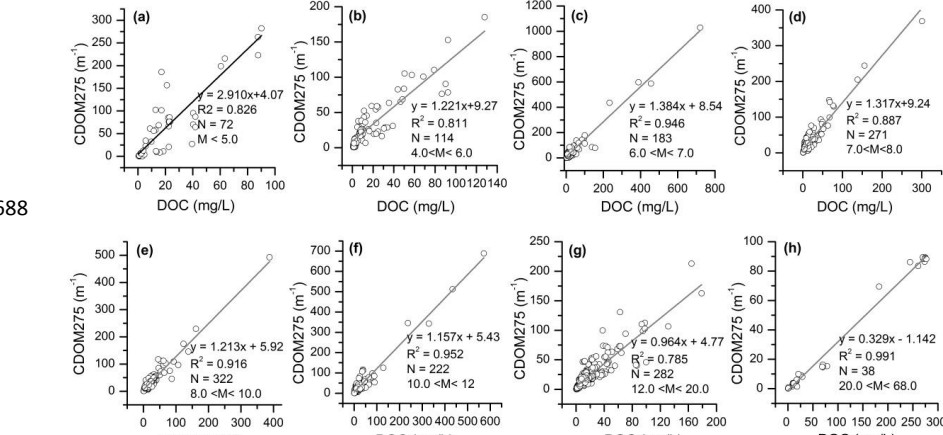







Fig.6. Relationship between CDOM 275 and DOC concentration. (a) regression
model with pooled dataset; (b) regression model with DOC concentration less than
300 mg/L; (c) regression model with natural logarithmic transformed data.

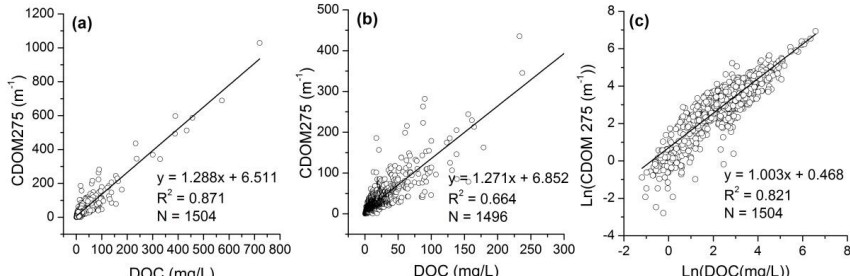






















**Tables**

Table 1 descriptive water quality characteristics of different types of waters

|  |  | DOC (mg/L) | EC µs/cm | pH | TP (mg/L) | TN (mg/L) | TSM (mg/L) | Chl-a (µg/L) |
|---|---|---|---|---|---|---|---|---|
| FW | Mean | 10.2 | 434.0 | 8.2 | 0.5 | 1.6 | 67.8 | 78.5 |
| | Range | 1.9-90.2 | 72.7-1181.5 | 6.9-9.3 | 0.01-10.4 | 0.001-9.5 | 0-1615 | 1.4-338.5 |
| SW | Mean | 27.3 | 4109.4 | 8.6 | 0.4 | 1.4 | 115.7 | 9.0 |
| | Range | 2.3-300.6 | 1067-41000 | 7.1-11.4 | 0.01-6.3 | 0.6-11.0 | 1.4-2188 | 0-113.7 |
| RW | Mean | 8.3 | 10489.1 | 7.8-9.5 | -- | -- | -- | -- |
| | Range | 0.9-90.2 | 3.7-1000 | 8.6 | -- | -- | -- | -- |
| UW | Mean | 19.44 | 525.4 | 8.0 | 3.4 | 3.5 | 50.5 | 38.9 |
| | Range | 3.5-123.3 | 28.6-1525 | 6.4-9.2 | 0.03-32.4 | 0.04-41.9 | 1-688 | 1.0-521.1 |
| WW | Mean | 67.0 | 1387.6 | 8.1 | 0.7 | 4.3 | 181.5 | 7.3 |
| | Range | 7.3-720 | 139-15080 | 7.0-9.7 | 0.1-4.8 | 0.5-48 | 9.0-2174 | 1.0-159.4 |
| IMW | Mean | 6.7 | 242.8 | 8.3 | 0.19 | 1.1 | 17.4 | 1.1 |
| | Range | 0.3-76.5 | 1.5-4350 | 6.7-10 | 0.02-2.9 | 0.3-8.6 | 0.3-254.6 | 0.28-5.8 |


Note: FW, fresh lake water; SW, saline lake water, RW, river or stream water; UW, urban water;
WW, winter water from Northeast China; IMW, ice melt water from Northeast China.




Table 2 descriptive statistics of dissolved organic carbon (DOC) and $a_{\text{CDOM}}(440)$ in various types of waters.

| Type | Region | DOC | | | | $a_{\text{CDOM}}(440)$ | | | |
|---|---|---|---|---|---|---|---|---|---|
| | | Min | Max | Mean | S.D | Min | Max | Mean | S.D |
| River | Liaohe | 3.6 | 48.2 | 14.3 | 9.49 | 0.46 | 3.68 | 0.92 | 0.58 |
| | Qinghai | 1.2 | 8.5 | 4.4 | 1.96 | 0.13 | 2.11 | 0.54 | 0.63 |
| | Inner M | 16.9 | 90.2 | 40.4 | 24.84 | 0.32 | 7.46 | 1.03 | 2.11 |
| | Songhua | 0.9 | 21.1 | 8.1 | 4.96 | 0.32 | 18.93 | 3.2 | 4.19 |
| Saline | Qinghai | 1.7 | 130.9 | 67.9 | 56.7 | 0.13 | 0.86 | 0.36 | 0.23 |
| | Hulunbir | 8.4 | 300.6 | 68.5 | 69.2 | 0.82 | 26.21 | 4.41 | 4.45 |
| | Xilinguo | 3.74 | 45.4 | 14.2 | 8.8 | 0.36 | 4.7 | 1.34 | 0.88 |
| | Songnen | 3.6 | 32.6 | 16.4 | 7.4 | 0.46 | 33.80 | 2.4 | 3.78 |