# Peer review of "A systematic examination of the relationship between CDOM and"

_Hydrology and Earth System Sciences, 2016_

## Referee Comment (RC1) · Anonymous Referee #1 · 8 Sep 2016

General Comments:

This study examines the linkage between CDOM and DOC in numerous types of water systems throughout China. As expected, this relationship is highly dependent on the type of environmental setting (e.g. river, lake, salt water, watershed types, etc). The study provides a large dataset that will be potentially very useful for future remote sensing studies in China. However, I would like to see more discussion of the broader relevance and how this fits into the global picture. For example, are the findings unique to Chinese water bodies or will these water body types have similar DOC/CDOM relationships globally or at least in similar latitudes?

There was very little discussion of findings in the literature with respect to the findings

of this study other than citing references to speculate on mechanisms. I would like to see more comparison of the slope results in the context of the above question. For example, do lakes generally fall within the same slope range?

Another major aspect that's lacking from this study, which focuses on inland waters, is the role of hydrology on these findings. It is surprising to not see any attempt to characterize the rivers studied in terms of average annual discharge and/or the season that samples were collected. For example, I am curious whether spatial or temporal variability plays a larger role in shaping the observations made here. At a minimum there needs to be discussion of the role that hydrology may play (there was one sentence saying hydrology wasn't considered).

Another point of discussion is whether or not these systems can feasibly be studied via remote sensing. There is no indication of the spatial scales that are being discussed. For example, a small stream cannot be resolved from space, so the authors should explain why it's important that we know a CDOM/DOC linkage for these waters. Perhaps because CDOM is easier/cheaper to analyze than DOC so we can get better temporal resolution in the future to address variability with the hydrograph? I think that Table 1 would be more informative if some basic information was provided about the size of the water bodies that are being examined. For example, adding the range/average of the basin scale and or river channel width/lake diameter to table 1 along with discharge if available would be useful. This would also allow some discussion of where the CDOM/DOC linkage is most robust and whether those particular water body types would be feasible for remote sensing applications (e.g. discuss the resolution of different satellites with respect to basin scale).

In its current form the manuscript does not provide much information that would be useful to the broader scientific community. However, it is a strong dataset, and if presented appropriately, this could have some very useful insight for the community. Finally, the authors should carefully review the manuscript for grammatical errors throughout (not all were noted), and potentially consider hiring a professional editor if unable to make

these corrections independently.

Specific Comments:

Line 47: This sentence doesn't seem to add much and the citation of Raymond et al (2013) doesn't really speak to what the article was about. This paper estimated that inland waters outgas 2.1 Pg C/yr, further constraining the global carbon budget.

Line 48: This would transition better if it was mentioned why remote sensing would be a useful tool, e.g. better spatiotemporal resolution.

Line 54: This sounds backwards. DOC is the larger pool, of which CDOM makes up a large fraction.

Line 54: DOC strictly speaking is net necessarily a "source of nutrients". If you said DOM, this could be true, but we typically think of things like N and P (organic or inorganic) as "nutrients." DOC is more aptly a "substrate" for heterotrophic bacteria, i.e. an energy source.

Line 55-56: This part of the sentence needs to be fixed for grammatical errors and clarity. Also, the Jaffe et al (2008) reference is not really relevant here. This paper looked at optical properties of CDOM, and any discussion of breakdown of allochthonous (i.e. terrestrial) DOC is inferential at best. Raymond et al (2013) also doesn't actually examine the mechanistic breakdown of DOC to CO2, they simply calculate global CO2 outgassing rates. The conclusion that this CO2 is from DOC breakdown comes from other studies. Some more suitable references that actually look at the breakdown of terrestrial DOC would be the following and references therein:

Ward, N.D., Keil, R.G., Medeiros, P.M., Brito, D.C., Cunha, A.C., Dittmar, T., Yager, P.L., Krusche, A.V. and Richey, J.E., 2013. Degradation of terrestrially derived macro-molecules in the Amazon River. Nature Geoscience, 6(7), pp.530-533.

Mayorga, E., Aufdenkampe, A.K., Masiello, C.A., Krusche, A.V., Hedges, J.I., Quay, P.D., Richey, J.E. and Brown, T.A., 2005. Young organic matter as a source of carbon

dioxide outgassing from Amazonian rivers. Nature,436(7050), pp.538-541.

Line 131: What pore-size, diameter and manufacturer were the filters? Was surface water collected for all of these analyses? What type of bottles were used and how were they cleaned? How were samples stored and for how long before analyses were performed? It seems as though bulk water was brought to the lab for filtration, so it's important to know how long samples sat and at what temperature. The decomposition processes alluded to in the introduction can occur quite rapidly. This appears to be noted on line 164…I would prefer this to be more upfront assuming the timing was the same for all bulk analyses. Also describe if samples were stored on ice, etc.

Line 134: It should at least be mentioned what type of filters were used.

Line 136: GF/F filters are typically 0.7um nominal pore size. Is this description accurate? Also, were samples preserved in any way?

Line 192: This raises an important point. This manuscript does not consider hydrographic variability in its discussion of CDOM/DOC, which are tightly coupled to river discharge. For example, one could see large variability at one particular site throughout the hydrograph and even hourly during rapid events. It's not clear what amount of the variability observed in this study is due to site differences versus hydrographic differences. This should at least be minimally addressed in the discussion. The next sentence begins to address this, but the odd phrasing for both sentences don't adequately get the point across.

Line 203: There should be some level of discussion of why DOC was so variable in each river, i.e. hydrologic controls. There needs to be some level of discussion on hydrologic controls. For example, DOC has been shown to be tightly linked to discharge both seasonally and especially during rapid storm event. See these articles and the references therein:

Raymond, P.A. and Saiers, J.E., 2010. Event controlled DOC export from forested

watersheds. Biogeochemistry, 100(1-3), pp.197-209.

Ward, N.D., Richey, J.E. and Keil, R.G., 2012. Temporal variation in river nutrient and dissolved lignin phenol concentrations and the impact of storm events on nutrient loading to Hood Canal, Washington, USA.Biogeochemistry, 111(1-3), pp.629-645.

Line 322: It's not clear how this conclusion was reached. Is it speculation based on literature?

Line 355-369: Are there comparisons to other studies in the literature that can be discussed/compared to here?

Line 372-405: Are there comparisons to other studies in the literature that can be discussed/compared to here?

Technical Corrections:

Line 17: Add "and" to end of list, also write "rivers and streams" if both were studied.

Line 18: Use the past tense, i.e. "ranged"

Line 20: It's not immediately obvious what is meant by "winter waters."

Line 23: "expected"

Line 24: Replace "sunshine" with "daylight." Remove "the" from "In the contrast"

Line 29: Make this sentence read more clearly.

Line 58-59: Fix grammatical errors (i.e. add "and")

Line 59: Choose more formal and grammatically correct wording than "A bunch of researches"

Line 69: Fix grammatical errors

Line 72: Fix grammatical errors

Line 74: Fix grammatical errors

Line 80: "Relatively"

Line 86: Consider re-wording this sentence

Line 89: "Two forms of carbons" is not accurate. CDOM is a subset of DOC.

Line 90-92: This is somewhat redundant as the same argument was made on lines 62-64.

Line 95: Fix grammatical errors

Line 103: Fix grammatical errors

Line 119: Use a word other than "data", i.e. dataset

Line 122: Fix grammatical errors (i.e. "of")

Line: 124-125: This sentence doesn't need to be included in the methods.

Results and Discussion:

Line 178: Perhaps add a subheading for "Bulk Geochemical Parameters" or something similar here.

Line 179: This is an odd sentence. The goal was to study diverse water types. It would be more fitting to say that geochemical properties across the unique water body types were diverse. Line 181: Use the past tense when describing results here and throughout the rest of the manuscript.

Line 185: Fix grammatical errors

Line 190: Fix grammatical errors

Line 212-218: Fix grammatical errors

Line 230: "wavelengths"

Line 243: Consider a different word than "endorsed."

Line 289: Fix grammatical errors

Line 303: Fix grammatical errors

Line 309-313: Fix grammatical errors

Line 315: Write (Chl-a = 7.3 ug/L). Also, this isn't an incredibly low value...that still indicates decent amounts of primary production.

Line 315: Fix grammatical errors

Line 317-319: Fix grammatical errors

Line 337: Fix grammatical errors

Line 377: Fix grammatical errors

Line 383-386: Fix grammatical errors

Line 397: Fix grammatical errors

Figure 1: There is probably better terminology than "Winter water", is this snow/ice-covered lakes, for example?

References:

The reference list appears to be incomplete. For example, Raymond et al (2013) is not present.

---

## Referee Comment (RC2) · Anonymous Referee #2 · 11 Oct 2016

GENERAL COMMENTS

This manuscript investigates the relationship between CDOM and DOC in a variety of inland water systems in China. The authors found, as has been shown before, that the predictive power of CDOM vs DOC concentration regressions vary; and that variation is likely associated to other biogeochemical factors. The data set in this study is extensive and representative of numerous types of water bodies, and has great potential to help inform organic carbon transport and dynamics for continental China; however, there is a lot of room for improvement. In general, the figures and tables seem appropriate. The introduction is vague and the results and discussion section is poor and many important results are overlooked or addressed superficially. The comparison of

regression slopes made among regions for the different types or water bodies is based on data that are not shown in any figures or tables, the regions used throughout the paper are not described in the site description. A deeper discussion of the potential mechanisms that drive CDOM chemistry and thus the DOC/CDOM correlations is strongly recommended for all the different water types investigated. The authors do a better job at proposing mechanisms for the results obtained for ice covered systems, however, that section is not entirely clear either and needs refining too. Another issue is the use of SUVA and E to separate the data, it would be really positive for the manuscript if the authors explained better the reasoning behind this kind of data sorting and compared their results to other people doing the same (as far as I could tell, this type of sorting has not been previously published). I believe there is a major flaw in the use of SUVA (see specific comments about section 3.3.1). Finally, the authors claim how important the investigation of correlations between CDOM and DOC are for feeding remote sensing models, however, they chose to quantify CDOM using the 275 nm wavelength while most empirical remote sensing methods used for inland waters are based on reflectance at wavelengths > 500 nm. A major concern is the poor grammar and lack of clarity in the text in numerous occasions throughout the manuscript, it is recommended that the authors revise this heavily to ensure no grammatical errors are present and ideas are clearly stated.

SPECIFIC COMMENTS (Note that grammatical errors are not being addressed/corrected)

Introduction:

Lines 46-47: Statement needs clarification.

Lines 55-56: Statement about mineralization needs rewording.

Line 70: "Gulf of Mexico" is the proper name.

Lines 87-92: This has already been mentioned in previous paragraphs of the introduction. Eliminate unnecessary repetition.

Line 89: CDOM and DOC are not "two forms of carbon". DOC is a component of the Earth's carbon pool; CDOM is a fraction of the natural organic matter pool, defined according to its optical properties and contains not only organic carbon but also nitrogen, phosphorus and sulfur.

Lines 95-98: I do not see a difference between objectives 1 and 2.

Lines 105-107: this idea has been mentioned once or twice already in the introduction; it would be better if the authors were more specific about how this study could inform remote sensing data for continental China, mentioning for example the data gaps or the limitations of previous studies. Also, prediction of DOC concentration from optical properties is not only useful for feeding remote sensing models. The authors could highlight other positive outcomes of this kind of analysis, especially for fluvial systems where the remote sensing techniques are more limited. It would be useful to add a paragraph about expected results to the end of the introduction.

Materials and methods:

Line 112: is the data set for freshwater lakes the same as the one used by Zhao et al in Biogeosciences, 13, 1635–1645, 2016? If so please indicate this and clarify that these results, although corresponding to the same data set do not represent previously published work.

Lines 124-125: this statement does not belong to this section; it needs to be eliminated or moved to the end of the introduction (see comment above about expected results).

Line 136: GF/F = glass fiber filters have a nominal pore size of 0.7 um. Please correct the pore size or the filter type.

Line 151: Samples for DOC where filtered through 0.7 (or 0.45 [not clear]) um filters according to Line 136, on the other hand, samples for CDOM were filtered through 0.22 um filters. How can this difference in sample treatment affect the results?

Line 156: This corresponds to the Napierian absorption coefficient, please specify this in the text.

Line 163: Zhang et al 2007 does not really explain the use of optical density over 740-750 nm as a correction factor for aCDOM, please cite a more appropriate article.

Lines 171-177: Why to describe the determination of the spectral slope if it is not presented in the results? This section is unnecessary.

A better description of the sampling locations or regions is needed, perhaps a table with detailed information about all water bodies sampled (location, dates, number of samples at each site, classification in this paper) in the supplementary material. Also, clarification is needed about how urban waters were classified as so; in other words, what parameter(s) was(were) used to define water bodies as urban?

Results and discussion:

Be consistent with the use of units, for example: do not mix ug/L with ug L-1. Also, there should be a space between a number and the unit in all cases, i.e., 10 mg not 10mg Streams and rivers are also freshwater systems (unless they are estuarine systems). It is confusing to use "freshwaters" to refer to freshwater lakes, please be specific if you are referring to lakes or streams/rivers, this applies to Figure 1 as well.

Line 194: this statement needs rewording. It is unintelligible.

Line 207: "inverse trend were" is not the appropriate wording, the authors are not describing a trend. A more suitable wording would be: "the opposite was found", or something along those lines.

Line 228: a more appropriate title would be "freshwater lakes and reservoirs"

Line 234-248: it is unclear how the authors make conclusions about DOC biogeochemistry in different regions of China (i.e., North China and East China; Northeast China, etc.) based on a regression analysis of the data set from different regions. Where are

these results presented? Where are these regions?

Lines 256-263: How were these slope values obtained? It would be useful to see the regressions for each of the regions that are mentioned in this section, and the slope values should be tabulated. Also each of the regions the authors are referring to should be shown either in Figure 1 or in a separate figure to clearly show where the regions are. This is related to the previous comment

Lines 266-267: This last statement is vague and gives the idea that the authors have also collected and analyzed remote sensing data. Please reword and focus.

Lines 247-280: See comment about lines 256-263.

Line 286: there are many other publications that are more appropriate citations for this statement than Jaffe et al 2008. For example: Williams et al 2010 L&O; Graeber et al 2012; etc. Also it would be interesting to examine in more detail how the variation in slopes compare to other results such as the ones published by Helms et al. and Spencer et al. and provide a more mechanistic explanation for this change in slope.

Line 294: provide citations for this statement.

Line 298-301: again, this conclusion is very vague, how exactly may urbanization affect the chemistry of dissolved organic matter in order to result in poor associations between DOC and CDOM? Is there literature showing similar results? What are the potential mechanisms?

Line 315: What does the number in parenthesis mean?

Line 322: Müller et al 2011 is not listed in the reference list.

Line 324: Stedmon et al 2009 is not listed in the reference list.

Lines 337-338: This statement is vague and unclear.

Lines 340-341: This is unclear, how can SUVA values "reflect" the regression slope for

DOC/CDOM.

Lines 341-345: The differences in SUVA suggested by the authors are not as clear as they indicate. A lot of overlapping exists in SUVA values across water types, it might be convenient to conduct an analysis of variance to determine significant differences across groups.

Section 3.3.1: As far I can tell, this kind of data sorting is a redundant exercise and it is obvious that better correlations than those obtained with the pooled data would be obtained if SUVA254 is used to sort the data. Most likely a254 and a275 are strongly correlated, thus, SUVA254 is pretty much the equivalent of the ratio of a275 to DOC which is what defines the slope of the a275 vs DOC regression, mathematically. So if SUVA254 is used to sort the data, you are practically grouping the samples that distribute more closely along a slope value. This can be easily seen in the slope values of each of the regressions of the sorted data in Figures 4a-h: the slope increases systematically as the SUVA254 range increases. I show this graphically in the attached Figure 1. I used the middle point of the different SUVA bins created by the authors (SUVA<1.0, 1.0<SUVA<2.0, 2.0<SUVA< 3.0, 3.0< SUVA< 4.0, 4.0<SUVA<5.0, 5.0<SUVA<6.0, 6.0<SUVA< 8.0, and 8.0< SUVA< 13.1), that is: 0.5, 1.5, 2.5, 3.5, 4.5, 5.5, 7, and 10.75, as a rough representation of the average SUVA254 value for each bin (y axis) and plotted it against the regression slope of each of the binned data sets (Fig 4a-h of manuscript). A clear linear correlation exemplifies the redundancy of using SUVA254 to sort the data. I strongly discourage this approach as a means to improve the correlations between CDOM and DOC.

Line 371: This heading should be 3.3.2

Section 3.3.2 (incorrectly named 3.3.1): see general comments.

FIGURE 4a: "greater than" symbol is incorrect, according to Line 539 it should be SUVA < 1. Correct also in the figure caption.

[Figure]

**REFERENCES**

It is suggested to read and incorporate the work by Brezonik et al al 2015 http://dx.doi.org/10.1016/j.rse.2014.04.033

[Figure]

[Figure]

**Fig. 1.** Middle point of SUVA bin vs regression slopes in sorted data.

---

## Author Comment (AC1) · 24 Nov 2016

Reviewer #1

General Comments: This study examines the linkage between CDOM and DOC in numerous types of water systems throughout China. As expected, this relationship is highly dependent on the type of environmental setting (e.g. river, lake, salt water, watershed types, etc). The study provides a large dataset that will be potentially very useful for future remote sensing studies in China. However, I would like to see more discussion of the broader relevance and how this fits into the global picture. For example, are the findings unique to Chinese water bodies or will these water body types have similar DOC/CDOM relationships globally or at least in similar latitudes? Response:

[Figure]

the authors thank for the valuable comments. As we know that the relationship between CDOM and DOC is depending on the complexity of chromophoric components of CDOM, which is directly linked with DOC. Waters from different source may have different chromophoric components, the molecular weights and the chromophoric fractions are also varied in different water sources, which ultimately affect the relationships between CDOM and DOC. CDOM and DOC sources have strong effects on the relationship, the other two major factors that affect the relationship are photo-oxidation and microbial degradation on the CDOM components. This investigation covers different water types, particularly these from Tibetan Plateau, which represent an extreme environmental condition except the Antarctica continent, and water samples from this plateau can be very valuable to examine the relationship between CDOM and DOC. In addition, urban waters generally represent CDOM and DOC with relatively complex conditions and the components for CDOM is much complicated to that from natural water bodies, thus which also provide valuable information for the systematic examination the relationship between CDOM and DOC. Considering the various water types examined in this study, it would represent most of the water types at global scale, and forms into the global picture for examining the relationship between CDOM and DOC. The authors added this discussion in the revised manuscript, thanks again for the comments.

There was very little discussion of findings in the literature with respect to the findings of this study other than citing references to speculate on mechanisms. I would like to see more comparison of the slope results in the context of the above question. For example, do lakes generally fall within the same slope range? Response: the authors agree with the comments, comparisons were made between the slopes between this study and literatures from previous studies. However, it also should be pointed out that different reference spectral bands were applied, thus make the comparison very hard.

Another major aspect that's lacking from this study, which focuses on inland waters, is the role of hydrology on these findings. It is surprising to not see any attempt to char-

acterize the rivers studied in terms of average annual discharge and/or the season that samples were collected. For example, I am curious whether spatial or temporal variability plays a larger role in shaping the observations made here. At a minimum there needs to be discussion of the role that hydrology may play (there was one sentence saying hydrology wasn't considered). Response: thanks for the comments. You are right that hydrology condition will definitely affect the relationship between CDOM and DOC. Originally we try to avoid this factor since a lot of factors need to be considered with respecting to different water types. Also most of the rivers only sampled once in our study, we have three sampling stations across the Songhua River, the Yalu River and the Hunjiang River, where time series samples were collected, these data were analyzed and provided in the revised manuscript. It also should pointed out that it is very hard to get hydrological data, and also the data is not released within two years, thus we did not collected current hydrological data in 2015 to examine the relationship between river flow and its impact on DOC and CDOM. Instead, we used some data we could collected over these rivers, and multi-year averaged flow data were used, that is the best we could do right now.

Another point of discussion is whether or not these systems can feasibly be studied via remote sensing. There is no indication of the spatial scales that are being discussed. For example, a small stream cannot be resolved from space, so the authors should explain why it's important that we know a CDOM/DOC linkage for these waters. Perhaps because CDOM is easier/cheaper to analyze than DOC so we can get better temporal resolution in the future to address variability with the hydrograph? I think that Table 1 would be more informative if some basic information was provided about the size of the water bodies that are being examined. For example, adding the range/average of the basin scale and or river channel width/lake diameter to table 1 along with discharge if available would be useful. This would also allow some discussion of where the CDOM/DOC linkage is most robust and whether those particular water body types would be feasible for remote sensing applications (e.g. discuss the resolution of different satellites with respect to basin scale). Response: the authors really thanks for

these valuable and helpful comments and suggestions. Your comments were adopted and incorporated in the revised manuscript, which is really helpful to strengthen the manuscript.

In its current form the manuscript does not provide much information that would be useful to the broader scientific community. However, it is a strong dataset, and if presented appropriately, this could have some very useful insight for the community. Finally, the authors should carefully review the manuscript for grammatical errors throughout (not all were noted), and potentially consider hiring a professional editor if unable to make these corrections independently. Response: the authors really thank for the comments, we took great efforts in interpretation of the findings from this large dataset, and tried best to reach useful information for the broader scientific community. The authors have carefully reviewed the manuscript, and a professional editor was used for the grammatical corrections.

Specific Comments: Line 47: This sentence doesn't seem to add much and the citation of Raymond et al (2013) doesn't really speak to what the article was about. This paper estimated that inland waters outgas 2.1 Pg C/yr, further constraining the global carbon budget. Response: the authors thank for the comments. What the authors try to say here is that the current studies have a rough idea about the role of inland water play in carbon cycling; however, how much of carbon stored in lakes, reservoirs and also rivers is not clear. The citation of Raymond et al. (2013) was removed and Tranvik et al. (2009) was added in the revised manuscript.

Line 48: This would transition better if it was mentioned why remote sensing would be a useful tool, e.g. better spatiotemporal resolution. Response: the authors thank for the comments, and your kind suggestion was adopted in the revised manuscript.

Line 54: This sounds backwards. DOC is the larger pool, of which CDOM makes up a large fraction. Response: the authors thank for the comments, this sentence was rephrased in the revised manuscript.

Line 54: DOC strictly speaking is net necessarily a "source of nutrients". If you said DOM, this could be true, but we typically think of things like N and P (organic or inorganic) as "nutrients." DOC is more aptly a "substrate" for heterotrophic bacteria, i.e. an energy source. Response: the authors really thank for the comments, this sentence was rephrased in the revised manuscript.

Line 55-56: This part of the sentence needs to be fixed for grammatical errors and clarity. Also, the Jaffe et al (2008) reference is not really relevant here. This paper looked at optical properties of CDOM, and any discussion of breakdown of allochthonous (i.e. terrestrial) DOC is inferential at best. Raymond et al (2013) also doesn't actually examine the mechanistic breakdown of DOC to CO2, they simply calculate global CO2 outgassing rates. The conclusion that this CO2 is from DOC breakdown comes from other studies. Some more suitable references that actually look at the breakdown of terrestrial DOC would be the following and references therein: Ward, N.D., Keil, R.G., Medeiros, P.M., Brito, D.C., Cunha, A.C., Dittmar, T., Yager, P.L., Krusche, A.V. and Richey, J.E., 2013. Degradation of terrestrially derived macromolecules in the Amazon River. Nature Geoscience, 6(7), pp.530-533. Mayorga, E., Aufdenkampe, A.K., Masiello, C.A., Krusche, A.V., Hedges, J.I., Quay, P.D., Richey, J.E. and Brown, T.A., 2005. Young organic matter as a source of carbon dioxide outgassing from Amazonian rivers. Nature, 436(7050), pp.538-541. Response: the authors really thank for the comments, these appropriate citations were added in the reference list in the revised manuscript.

Line 131: What pore-size, diameter and manufacturer were the filters? Was surface water collected for all of these analyses? What type of bottles were used and how were they cleaned? How were samples stored and for how long before analyses were performed? It seems as though bulk water was brought to the lab for filtration, so it's important to know how long samples sat and at what temperature. The decomposition processes alluded to in the introduction can occur quite rapidly. This appears to be noted on line 164: : :I would prefer this to be more upfront assuming the timing was the

same for all bulk analyses. Also describe if samples were stored on ice, etc. Response: the authors really thank for the concerns and valuable comments. These descriptions about water sampling, samples keeping and shipping, and preprocessing were added in the revised manuscript.

Line 134: It should at least be mentioned what type of filters were used. Response: the authors really thank for the concerns, the type of the filters were added in the revised manuscript.

Line 136: GF/F filters are typically 0.7um nominal pore size. Is this description accurate? Also, were samples preserved in any way? Response: the authors thank for the concerns, we checked with the technician and the correct brand of filters were used in the revised manuscript.

Line 192: This raises an important point. This manuscript does not consider hydrographic variability in its discussion of CDOM/DOC, which are tightly coupled to river discharge. For example, one could see large variability at one particular site throughout the hydrograph and even hourly during rapid events. It's not clear what amount of the variability observed in this study is due to site differences versus hydrographic differences. This should at least be minimally addressed in the discussion. The next sentence begins to address this, but the odd phrasing for both sentences don't adequately get the point across. Response: the authors thank for the concerns, as mentioned in the previous responses,

Line 203: There should be some level of discussion of why DOC was so variable in each river, i.e. hydrologic controls. There needs to be some level of discussion on hydrologic controls. For example, DOC has been shown to be tightly linked to discharge both seasonally and especially during rapid storm event. See these articles and the references therein: Raymond, P.A. and Saiers, J.E., 2010. Event controlled DOC export from forested watersheds. Biogeochemistry, 100(1-3), pp.197-209. Ward, N.D., Richey, J.E. and Keil, R.G., 2012. Temporal variation in river nutrient and dissolved lignin phenol concentrations and the impact of storm events on nutrient loading to Hood Canal, Washington, USA.Biogeochemistry, 111(1-3), pp.629-645. Response: the authors really thank for the concerns. Some discussion on the variability of the DOC in different rivers, and also its connection with hydrology, please see the details in the revised manuscript. As mentioned in one of the responses to the reviewer's comments, we did not pay too much attention to the seasonal variation of DOC and CDOM from rivers since most of the samples from rivers were collected just once in this study. However, we do collect time series samples from Songhua River and Yalu Rive in Northeast China, some of the results were added in the revised manuscript. Line 322: It's not clear how this conclusion was reached. Is it speculation based on literature? Response: the authors thank for the concern, as you guessed, this conclusion was mainly based on literature. However, our own data set (which was not shown in the first version of the manuscript) also support this conclusion, and was added in the revised manuscript.

Line 355-369: Are there comparisons to other studies in the literature that can be discussed/compared to here? Response: the authors thank for the concern, as you may know that the second reviewer strongly suggested that this paragraph should be removed since the classification of CDOM based on SUVA254 is not meaningful, thus comparison was not conducted in the revised manuscript.

Line 372-405: Are there comparisons to other studies in the literature that can be discussed/compared to here? Response: the authors really thank for the comments. Actually, this is the first study tries to establish the relationships between CDOM and DOC based CDOM absorption grouping, however, comparisons were still made and details can be tracked in the revised manuscript.

Technical Corrections: Line 17: Add "and" to end of list, also write "rivers and streams" if both were studied. Response: we corrected the sentence as suggested, many thanks. Line 18: Use the past tense, i.e. "ranged" Response: thank you, the tense was corrected as suggested. Line 20: It's not immediately obvious what is meant by

"winter waters." Response: thanks, the authors change "winter waters" to "ice-covered waters". Line 23: "expected" Response: thanks, it has been corrected. Line 24: Replace "sunshine" with "daylight." Remove "the" from "In the contrast" Response: the authors thank for the corrections. Line 29: Make this sentence read more clearly. Response: the sentence is rephrased; please check it out in the revised manuscript. Line 58-59: Fix grammatical errors (i.e. add "and") Response: the authors thank for the suggestion, and the grammatical errors were fixed in the revised manuscript. Line 59: Choose more formal and grammatically correct wording than "A bunch of researches" Response: thanks for the suggestion, we used "numerous". Line 69: Fix grammatical errors Response: thanks a lot, the grammatical error was corrected. Line 72: Fix grammatical errors Response: thanks a lot, the grammatical error was corrected. Line 74: Fix grammatical errors Response: thanks a lot, the grammatical error was corrected. Line 80: "Relatively" Response: the authors thank for pointing out the wrong word, it was corrected. Line 86: Consider re-wording this sentence Response: the authors thank for the suggestion, this sentence was rephrased in the revised manuscript. Line 89: "Two forms of carbons" is not accurate. CDOM is a subset of DOC. Response: the authors corrected the sentence with accurate choosing of word. Line 90-92: This is somewhat redundant as the same argument was made on lines 62-64. Response: the authors thank for the suggestion, and the full sentence was removed. Line 95: Fix grammatical errors Response: thanks for the suggestion, the grammatical error was fixed. Line 103: Fix grammatical errors Response: thanks, the grammatical error was fixed in revised manuscript. Line 119: Use a word other than "data", i.e. dataset Response: your kind suggestion was adopted in revised manuscript. Line 122: Fix grammatical errors (i.e. "of") Response: the authors thank for the suggestion, and the sentence was rephrased in the revised manuscript. Line: 124-125: This sentence doesn't need to be included in the methods. Response: the authors agree with the reviewer's suggestion, and the sentence was removed in the revised manuscript.

Results and Discussion: Line 178: Perhaps add a subheading for "Bulk Geochemical Parameters" or something similar here. Response: the authors thank for the suggestion, we added a subheading of "bulk geochemical parameters characteristics" in the revised manuscript, and we really appreciated for the valuable suggestion. Line 179: This is an odd sentence. The goal was to study diverse water types. It would be more fitting to say that geochemical properties across the unique water body types were diverse. Response: the authors thank for the suggestion, and the sentence was rephrased. Line 181: Use the past tense when describing results here and throughout the rest of the manuscript. Response: the authors thank for the suggestion, and the past tense was used for describing results throughout the rest of the manuscript. Line 185: Fix grammatical errors Response: thanks for the suggestion, and the grammatical error was fixed. Line 190: Fix grammatical errors Response: thanks for the suggestion, and the grammatical error was fixed. Line 212-218: Fix grammatical errors Response: thanks for the suggestion, and the grammatical error was fixed. Line 230: "wavelengths" Response: thanks for the correction, and the plural form was used. Line 243: Consider a different word than "endorsed." Response: thanks for the suggestion, and the sentence was reworded. Line 289: Fix grammatical errors Response: thanks for the suggestion, and the grammatical error was fixed. Line 303: Fix grammatical errors Response: thanks for the suggestion, and the grammatical error was fixed. Line 309-313: Fix grammatical errors Response: thanks for the suggestion, and the grammatical error was fixed. Line 315: Write (Chl-a = 7.3 ug/L). Also, this isn't an incredibly low value: : :that still indicates decent amounts of primary production. Response: thanks for the suggestion, the authors adopted the suggestion in the revised manuscript, further, the sentence was rephrased to achieve an accurate description. Line 315: Fix grammatical errors Response: thanks for the suggestion, and the grammatical error was fixed. Line 317-319: Fix grammatical errors Response: The grammatical errors were fixed, thanks for the suggestion. Line 337: Fix grammatical errors Response: The grammatical error was fixed, thanks a lot. Line 377: Fix grammatical errors Response: thanks for the suggestion, and the grammatical error was fixed. Line 383-386: Fix grammatical errors Response: the authors thank for the comment, and the grammatical errors were fixed in the revised manuscript.

Line 397: Fix grammatical errors Response: the authors thank for the suggestion, and the grammatical error was fixed.

Figure 1: There is probably better terminology than "Winter water", is this snow/icecovered lakes, for example? Response: thanks a lot for the suggestion, we reproduce figure 1 as suggested. References: The reference list appears to be incomplete. For example, Raymond et al (2013) is not present. Response: the cited paper was added in the reference list, thanks a lot.

[Figure]

---

## Author Comment (AC2) · 24 Nov 2016

Reviewer #2

GENERAL COMMENTS This manuscript investigates the relationship between CDOM and DOC in a variety of inland water systems in China. The authors found, as has been shown before, that the predictive power of CDOM vs DOC concentration regressions vary; and that variation is likely associated to other biogeochemical factors. The data set in this study is extensive and representative of numerous types of water bodies, and has great potential to help inform organic carbon transport and dynamics for continental China; however, there is a lot of room for improvement. In general, the figures and tables seem appropriate. The introduction is vague and the results and discussion

section is poor and many important results are overlooked or addressed superficially. The comparison of regression slopes made among regions for the different types or water bodies is based on data that are not shown in any figures or tables, the regions used throughout the paper are not described in the site description. A deeper discussion of the potential mechanisms that drive CDOM chemistry and thus the DOC/CDOM correlations is strongly recommended for all the different water types investigated. The authors do a better job at proposing mechanisms for the results obtained for ice covered systems, however, that section is not entirely clear either and needs refining too. Another issue is the use of SUVA and E to separate the data, it would be really positive for the manuscript if the authors explained better the reasoning behind this kind of data sorting and compared their results to other people doing the same (as far as I could tell, this type of sorting has not been previously published). I believe there is a major flaw in the use of SUVA (see specific comments about section 3.3.1). Finally, the authors claim how important the investigation of correlations between CDOM and DOC are for feeding remote sensing models, however, they chose to quantify CDOM using the 275 nm wavelength while most empirical remote sensing methods used for inland waters are based on reflectance at wavelengths > 500 nm. A major concern is the poor grammar and lack of clarity in the text in numerous occasions throughout the manuscript, it is recommended that the authors revise this heavily to ensure no grammatical errors are present and ideas are clearly stated. Responses: the authors really thank for the insightful comments, as you may see in the revised manuscript, we addressed most of the questions you raised in the review. We deleted the section 3.3.1, i.e., the regression based on SUVA254. Also, we added a figure to deal with the relationship between DOC and aCDOM(440), which could be used for remote sensing of CDOM or DOC since that is the most reference spectral band in this community.

SPECIFIC COMMENTS (Note that grammatical errors are not being addressed/corrected)

Introduction: Lines 46-47: Statement needs clarification. Response: thanks for the

comments; the authors have rephrased the sentence to achieve a clarification of the statement. Lines 55-56: Statement about mineralization needs rewording. Response: thanks for the comments; the sentence is rephrased. Line 70: "Gulf of Mexico" is the proper name. Response: thanks for the suggestion; the suggestion has been incorporated in the revised manuscript. Lines 87-92: This has already been mentioned in previous paragraphs of the introduction. Eliminate unnecessary repetition. Response: thanks for the comments; this part was removed in the revised manuscript. Line 89: CDOM and DOC are not "two forms of carbon". DOC is a component of the Earth's carbon pool; CDOM is a fraction of the natural organic matter pool, defined according to its optical properties and contains not only organic carbon but also nitrogen, phosphorus and sulfur. Response: Thanks for the comments, which help the authors to understand the difference and link between CDOM and DOC. However, this sentence was removed in the revised manuscript to responding to the previous comment. Lines 95-98: I do not see a difference between objectives 1 and 2. Response: Thanks for the comments; it is true that objectives 1 and 2 overlap, thus the second one is removed in the revised manuscript. Lines 105-107: this idea has been mentioned once or twice already in the introduction; it would be better if the authors were more specific about how this study could inform remote sensing data for continental China, mentioning for example the data gaps or the limitations of previous studies. Also, prediction of DOC concentration from optical properties is not only useful for feeding remote sensing models. The authors could highlight other positive outcomes of this kind of analysis, especially for fluvial systems where the remote sensing techniques are more limited. It would be useful to add a paragraph about expected results to the end of the introduction. Response: The authors really thank for the very valuable comments; several sentences about the expected results of the research is added in the revised manuscript.

Materials and methods: Line 112: is the data set for freshwater lakes the same as the one used by Zhao et al in Biogeosciences, 13, 1635–1645, 2016? If so please indicate this and clarify that these results, although corresponding to the same data set do not represent previously published work. Response: the authors thank for the

concerns. The data set for freshwater lakes is collected across China, e.g., the data set covers lakes at national scale, while the data set used in Zhao et al., in Biogosciences, 13, pp 1635-1645, 2016, was sampled only in a few lakes in Northeast China (see sampling locations in Figure 1 of this manuscript, and also the study area in Zhao et al., 2016 for details). Although some of the data used in Zhao was used in this study, it is only a small of part of the data set used in this study, and also the looking angle is different from Zhao et al. (2016), where fluorescence feature from CDOM is more concerned. Lines 124-125: this statement does not belong to this section; it needs to be eliminated or moved to the end of the introduction (see comment above about expected results). Response: the authors thank for the suggestion, and this sentence is removed in the revised manuscript. Line 136: GF/F = glass fiber filters have a nominal pore size of 0.7 um. Please correct the pore size or the filter type. Response: the authors thank for the comments, it turned out that we used Whatman cellulose acetate filter with pore size of 0.45 $\mu$m, and the information was corrected in the revised manuscript. Line 151: Samples for DOC where filtered through 0.7 (or 0.45 [not clear]) um filters according to Line 136, on the other hand, samples for CDOM were filtered through 0.22 um filters. How can this difference in sample treatment affect the results? Response: the authors thank for the concern. The standard protocol for DOC determination is generally filtered through 0.7 (early study applied this pore size) or 0.45 (nowadays applied this pore size); however, samples for CDOM absorption determination is generally filtered 0.22 um filters to avoid fine particle that might have scattering effect on CDOM absorption spectra (Babin et al., 2003). We did not think too much about this issue in the previous work, we presume that the filter pore size may have but very minor effect on the relationship between DOC and CDOM. Also, in our study, all the samples were pre-process with the same methods, thus the sample treatment should affect the result in very limited manner.

Line 156: This corresponds to the Napierian absorption coefficient, please specify this in the text. Response: the authors thank for the comment, we added the information in the revised manuscript as suggested. Line 163: Zhang et al 2007 does not really

explain the use of optical density over 740-750 nm as a correction factor for aCDOM, please cite a more appropriate article. Response: the authors thank for the comment, we cited the right paper (Babin et al., 2003) in the revised manuscript as suggested. Lines 171-177: Why to describe the determination of the spectral slope if it is not presented in the results? This section is unnecessary. Response: the authors really thank for the comments; this section was removed in the revised manuscript.

A better description of the sampling locations or regions is needed, perhaps a table with detailed information about all water bodies sampled (location, dates, number of samples at each site, classification in this paper) in the supplementary material. Also, clarification is needed about how urban waters were classified as so; in other words, what parameter(s) was (were) used to define water bodies as urban? Response: the authors thank for the valuable comments; a supplementary table was produced in the revised manuscript, and the information mentioned in the comments were provided in the supplementary table, please check the table in the supplementary material. Here, we would like to point out that the information for rivers or stream was not listed in the supplementary table due to the multi sampling points were collected along rivers or stream, thus the reader can reference the relative position through Figure 1. The definition of urban waters were added in the main text, please check it out in the revised manuscript.

Results and discussion: Be consistent with the use of units, for example: do not mix ug/L with ug L-1. Also, there should be a space between a number and the unit in all cases, i.e., 10 mg not 10mg Streams and rivers are also freshwater systems (unless they are estuarine systems). Response: thanks for the comments; the authors have checked these issues throughout the manuscript, and corrections were made correspondingly. Thanks again for the suggestions. It is confusing to use "freshwaters" to refer to freshwater lakes, please be specific if you are referring to lakes or streams/rivers, this applies to Figure 1 as well. Response: thanks for the comments; your recommendation was adopted in the revised manuscript, the authors have also

checked out the whole manuscript to clarify these confusions. Line 194: this statement needs rewording. It is unintelligible. Response: thanks for the suggestion; the authors rephrased this sentence in the revised manuscript to make it clear. Line 207: "inverse trend were" is not the appropriate wording, the authors are not describing a trend. A more suitable wording would be: "the opposite was found", or something along those lines. Response: the authors thank for the comment, and the suggestion was adopted in the revised manuscript. Line 228: a more appropriate title would be "freshwater lakes and reservoirs" Response: the authors really thank for the suggestion, and the title was modified as suggested. Line 234-248: it is unclear how the authors make conclusions about DOC biogeochemistry in different regions of China (i.e., North China and East China; Northeast China, etc.) based on a regression analysis of the data set from different regions. Where are these results presented? Where are these regions? Response: the authors really thank for the comments, a table with these related information was added in the revised manuscript, please see the details in the revised manuscript. Lines 256-263: How were these slope values obtained? It would be useful to see the regressions for each of the regions that are mentioned in this section, and the slope values should be tabulated. Also each of the regions the authors are referring to should be shown either in Figure 1 or in a separate figure to clearly show where the regions are. This is related to the previous comment. Response: the authors really thank for the comments, a table with these related information was added in the revised manuscript, please see the details in the revised manuscript. Lines 266-267: This last statement is vague and gives the idea that the authors have also collected and analyzed remote sensing data. Please reword and focus. Response: the authors really thank for the suggestion, and the title was modified as suggested. Lines 247-280: See comment about lines 256-263. Response: thanks for the valuable comments, as mentioned in the previous responses to the comments, a table with these information was added in the revised manuscript, please see the details in the added table. Line 286: there are many other publications that are more appropriate citations for this statement than Jaffe et al 2008. For example: Williams et al 2010 L&O; Grae-

ber et al 2012; etc. Also it would be interesting to examine in more detail how the variation in slopes compare to other results such as the ones published by Helms et al. and Spencer et al. and provide a more mechanistic explanation for this change in slope. Response: the authors really thank for the valuable comments, and more appropriate citations were added in the main text, also more detailed comparisons were added in the revised manuscript. Line 294: provide citations for this statement. Response: thanks for the comments; citations were added in the revised manuscript. Line 298-301: again, this conclusion is very vague, how exactly may urbanization affect the chemistry of dissolved organic matter in order to result in poor associations between DOC and CDOM? Is there literature showing similar results? What are the potential mechanisms? Response: thanks for the concern; additional statements and literature were provided in the revised manuscript. Line 315: What does the number in parenthesis mean? Response: thanks for the concern; the number in the parenthesis is chlorophyll-a concentration, the information was provided in the revised manuscript. Line 322: Müller et al 2011 is not listed in the reference list. Response: thanks for pointing out the error, this reference is replace by Zhang et al., 2010, and also added into the reference list in the revised manuscript. Line 324: Stedmon et al 2009 is not listed in the reference list. Response: thanks for the comment, and alternative cited literature was added in the revised manuscript. Lines 337-338: This statement is vague and unclear. Response: the authors thank for the comment, this statement was rephrased in the revised manuscript.

Lines 340-341: This is unclear, how can SUVA values "reflect" the regression slope for Lines 341-345: The differences in SUVA suggested by the authors are not as clear as they indicate. A lot of overlapping exists in SUVA values across water types, it might be convenient to conduct an analysis of variance to determine significant differences across groups. Response: the authors really thank for the valuable comment, this paragraph was removed, please see the revised manuscript for the details of revision.

Section 3.3.1: As far I can tell, this kind of data sorting is a redundant exercise and it

is obvious that better correlations than those obtained with the pooled data would be obtained if SUVA254 is used to sort the data. Most likely a254 and a275 are strongly correlated, thus, SUVA254 is pretty much the equivalent of the ratio of a275 to DOC which is what defines the slope of the a275 vs DOC regression, mathematically. So if SUVA254 is used to sort the data, you are practically grouping the samples that distribute more closely along a slope value. This can be easily seen in the slope values of each of the regressions of the sorted data in Figures 4a-h: the slope increases systematically as the SUVA254 range increases. I show this graphically in the attached Figure 1. I used the middle point of the different SUVA bins created by the authors (SUVA<1.0, 1.0<SUVA<2.0, 2.0<SUVA< 3.0, 3.0< SUVA< 4.0, 4.0<SUVA<5.0, 5.0<SUVA<6.0, 6.0<SUVA< 8.0, and 8.0< SUVA< 13.1), that is: 0.5, 1.5, 2.5, 3.5, 4.5, 5.5, 7, and 10.75, as a rough representation of the average SUVA254 value for each bin (y axis) and plotted it against the regression slope of each of the binned data sets (Fig 4a-h of manuscript). A clear linear correlation exemplifies the redundancy of using SUVA254 to sort the data. I strongly discourage this approach as a means to improve the correlations between CDOM and DOC. Response: the authors really thank for the comments; this paragraph was removed in the revised manuscript. Line 371: This heading should be 3.3.2 Response: thanks for the comments; series number for this subheading is corrected. Section 3.3.2 (incorrectly named 3.3.1): see general comments. Response: the authors thank for the comments, correction was made in the revised manuscript. FIGURE 4a: "greater than" symbol is incorrect, according to Line 539 it should be SUVA < 1. Correct also in the figure caption. Response: thanks for the comments.

REFERENCES It is suggested to read and incorporate the work by Brezonik et al al 2015 http://dx.doi.org/10.1016/j.rse.2014.04.033 Response: thanks for the comments; the suggested literature is incorporated in the revised manuscript, and has been added in the reference list.